# Reinforcement Learning Models of Human Behavior: Reward Processing in Mental Disorders

**Baihan Lin**
Columbia University
baihan.lin@columbia.edu

**Guillermo Cecchi**
IBM Research
gcecchi@us.ibm.com

**Djallel Bouneffouf**
IBM Research
djallel.bouneffouf@ibm.com

**Jenna Reinen**
IBM Research
jenna.reinen@ibm.com

**Irina Rish**
IBM Research
rish@us.ibm.com

## Abstract

Drawing an inspiration from behavioral studies of human decision making, we propose here a general parametric framework for a reinforcement learning (RL) problem, which extends the standard Q-learning approach to incorporate a two-stream framework of reward processing with biases biologically associated with several neurological and psychiatric conditions, including Parkinson's and Alzheimer's diseases, attention-deficit/hyperactivity disorder (ADHD), addiction, and chronic pain. The development of agents that react differently to different types of rewards can enable us to understand a wide spectrum of multi-agent interactions in complex real-world socioeconomic systems. Empirically, the proposed model outperforms Q-Learning (QL) and Double Q-Learning (DQL) in scenarios with certain reward distributions and real-world human decision making gambling tasks. On behavioral modeling, our parametric framework can be viewed as a first step towards a unifying computational model capturing reward processing abnormalities across multiple mental conditions and user preferences in recommendation systems.

To better understand and model human decision making, scientists have investigated reward processing mechanisms in healthy subjects [1]. However, neurodegenerative and psychiatric disorders associated with reward processing disruptions can provide an additional resource for deeper understanding of human decision making mechanisms. Various mental disorders, including depression, ADHD, addiction and even schizophrenia can be considered as "extreme points" in a continuous spectrum of behaviors and traits developed for various developmental and environmental adaptive purposes. Thus, modeling decision-making biases and traits associated with disorders may enrich the existing computational decision-making framework, leading to potentially more flexible and better-performing algorithms. Herein, we focus on reward-processing biases associated with several mental disorders, including Parkinson's and Alzheimer's disease, ADHD, addiction, and chronic pain. Is it possible to extend standard reinforcement learning algorithms to mimic human behavior in such disorders? Can such generalized approaches outperform standard RL algorithms on specific tasks?

We show that both questions can be answered positively. We build upon the QL, a state-of-art approach to RL problem, and extend it to a parametric version which allows to split the reward information into positive and negative streams with various reward-processing biases associated with particular disorders. For example, Parkinson's patients appear to learn better from negative rather than from positive rewards [2]; addictive behaviors are associated with an inability to forget strong stimulus-response associations from the past, i.e. to properly discount past rewards [3], and so on. More specifically, we propose a parametric model which introduces weights on incoming positive and negative rewards, and on reward histories, extending the standard parameter update rules in QL; tuning the parameter settings allows us to better capture specific reward-processing biases.

33rd Conference on Neural Information Processing Systems (NeurIPS 2019), Vancouver, Canada.

# 1 Neuroscience Motivation

**Neural representation of reward.** Evidence has linked dopamine function to reinforcement learning via midbrain neurons and connections to the basal ganglia, limbic regions, and cortex. Neuron firing rates computationally represent reward magnitude, expectancy, and violations (prediction error) and other value-based signals [4], allowing an animal to update and maintain value expectations associated with particular states and actions. When functioning properly, this helps an animal develop a policy to maximize outcomes by approaching/choosing cues with higher expected value and avoiding cues associated with loss or punishment. This is similar to reinforcement learning widely used in computing and robotics [5], suggesting mechanistic overlap in humans and AI. Evidence of Q-learning and actor-critic models have been observed in spiking activity in midbrain dopamine neurons in primates [6] and in the human striatum using the BOLD signal [7].

**Positive vs. negative learning signals.** Phasic dopamine signaling represents bidirectional (positive and negative) coding for prediction error signals [8], but underlying mechanisms show differentiation for reward relative to punishment learning [9]. Though representation of cellular-level aversive error signaling has been debated [10], it is widely thought that rewarding, salient information is represented by phasic dopamine signals, whereas reward omission or punishment signals are represented by dips or pauses in baseline dopamine firing [4]. These mechanisms have downstream effects on approach behavior and action selection, and are reflected in neural direct and indirect pathways, generating opposing outputs that either facilitate or inhibit action [11]. Manipulating these circuits through pharmacological measures and disease has supported the idea that learning mechanisms differ when the prediction error is positive or negative [2], and elucidate our understanding of loss/gain decision biases in humans [12].

**Clinical Implications.** A body of recent literature has demonstrated that a spectrum of neurological and psychiatric disease symptoms are related to biases in learning from positive and negative feedback [13]. Studies in humans have shown that when reward signaling in the direct pathway is over-expressed, this may enhance state value and incur pathological reward-seeking behavior, like gambling or substance use. Conversely, enhanced aversive error signals result in dampened reward experience thereby causing symptoms like apathy, social withdrawal, fatigue, and depression. Both genetic predispositions and experiences during critical periods of development can predispose an individual to learn from positive or negative outcomes, making them more or less at risk for brain-based illnesses [14]. This highlight our need to understand how intelligent systems learn from rewards and punishments, and how experience sampling may impact reinforcement learning during influential training periods.

# 2 Human Q-Learning

We will now introduce a more general formulation of Q-Learning incorporating the reward signals from a positive and a negative stream. We propose Human Q-Learning (HQL), outlined in Algorithm 1, which updates the Q values using four weight parameters: $\phi_1$ and $\phi_3$ are the weights of the previously accumulated positive and negative rewards, respectively, while $\phi_2$ and $\phi_4$ represent the weights on the positive and negative rewards at the current iteration. In our algorithm, we have two Q tables that we are using $Q^+$ and $Q^-$ which respectively record the positive and negative feedback. The parameter settings are summarized in Appendix C, where we use list our models associated with specific disorders.

---
**Algorithm 1** Human Q-Learning (HQL)

---
1: **For** each episode t **do**
2:     Initialize s
3:     **Repeat**
4:         $Q(s,a) := \phi_2 Q^+(s,a) + \phi_4 Q^-(s,a)$
5:         action $i_t = \arg \max_i Q_i(t)$, observe $s' \in S$, $r^+$ and $r^- \in R(s)$
6:         $Q^+(s,a) := \phi_1 \hat{Q}^+(s,a) + \alpha_t(r^+ + \gamma \quad max_{a'}\hat{Q}^+(s',a') - \hat{Q}^+(s,a))$
7:         $Q^-(s,a) := \phi_3 \hat{Q}^-(s,a) + \alpha_t(r^- + \gamma \quad max_{a'}\hat{Q}^-(s',a') - \hat{Q}^-(s,a))$
8:     **until** s is terminal

---

# 3 Empirical Results

Empirically, we evaluated the algorithms in two settings: the gambling game of a simple Markov Decision Process (MDP) and a real-life Iowa Gambling Task (IGT) [15]. To evaluate the performances of the algorithms, we applied the sports team scoring system as a scenario-independent measure.

**MDP example.** In this simple MDP example, a player starts from initial state A, choose between two actions: go left to reach state B, or go right to reach state C. Both states B and C reveals a zero rewards. From state B, the player has only one action to reach state D which reveals $n$ draws of rewards from a distribution $R_D$. From state C, the player has only one action to reach state E which reveals $n$ draws of rewards from a distribution $R_E$. The reward distributions of states D and E are both multimodal distributions (for instance, the reward $r$ can be drawn from a bi-modal distribution of two normal distributions $N(\mu = 10, \sigma = 5)$ with probability $p = 0.3$ and $N(\mu = -5, \sigma = 1)$ with $p = 0.7$). In the simulations, $n$ is set to be 50. The left action (go to state B) by default is set to have an expected payout lower than the right action. However, the reward distributions can be spread across both the positive and negative domains. For HQL, the reward is separated into a positive stream (if the revealed reward is positive) and a negative stream (if the revealed reward is negative).

Figure 1 shows an example scenario where the reward distributions, percentage of choosing the better action (go right), cumulative rewards and the changes of two Q-tables over the number of iterations, drawn with standard errors over 100 runs. Each trial consisted of a synchronous update of all 500 actions. With polynomial learning rates, we see Human Q-learning converges much more quickly than Q-Learning. To better evaluate the robustness of the algorithms, we simulated 100 randomly generated scenarios of bi-modal distributions, where the reward distributions can be drawn from two normal distribution with means as random integers uniformly drawn from -100 to 100, standard deviations as random integers uniformly drawn from 0 to 20, and sampling distribution $p$ uniformly drawn from 0 to 1 (assigning $p$ to one normal distribution and $1 - p$ to the other one). Each scenario was repeated 100 times. Table 1 summarizes the pairwise comparisons between Q-Learning (QL), Double Q-Learning (DQL) [16], Standard Human Q-Learning (SQL), Positive Q-Learning (PQL) and Negative Q-Learning (NQL), with the row labels as the algorithm X and column labels as algorithm Y giving $n : m$ in each cell denoting X beats Y $n$ times and Y beats X $m$ times. Among the five algorithms, SQL never fail catastrophically by maintaining an overall advantages over the others (with the highest average winning percentage of 0.68 while all others below 0.50). HQL seems to benefit from the sensitivity to two streams instead of collapsing them into the mean as in Q-Learning.

To explore the variants of HQL representing different mental disorders, we also performed the same experiments on the 7 disease models proposed in section **??**. Table 1 summarizes their pairwise comparisons with SQL, DQL and QL, where the average wins are computed averaged against three standard baseline models. Overall, PD ("Parkinson's"), CP ("Chronic Pain") and M ("moderate") performs relatively well when in this environments. With the same algorithmic framework as the mental agents, the standard HQL (SQL) can distinguish against most mental agents with the largest marginals (0.81 chance of beating a certain mental agents, while DQL with 0.65 and QL with 0.58). The variation of behaviors also suggest the proposed framework can potentially cover a wide spectrum of behavior by simply tuning the four hyperparameters.

**Iowa Gambling Task**. The original Iowa Gambling Task (IGT) studies decision making where the participant needs to choose one out of four card decks (named A, B, C, and D), and can win or lose money with each card when choosing a deck to draw from [17], over around 100 actions. In each round, the participants receives feedback about the win (the money he/she wins), the loss (the money he/she loses), and the combined gain (win minus lose). In the IGT setup, from initial state I, the player select one of the four deck to go to state A, B, C, or D, and reveals positive reward $r^+$ (the win), negative reward $r^-$ (the loss) and combined reward $r = r^+ + r^-$ simultaneously. Decks A and B by default is set to have an expected payout (-25) lower than the better decks, C and D (+25). For QL and DQL, the combined reward $r$ is used to update the agents. For HQL, PQL and NQL, the positive and negative streams are fed and learned independently given the $r^+$ and $r^-$. There are two major payoff schemes in IGT. In the traditional payoff scheme, the net outcome of every 10 cards from the bad decks (i.e., decks A and B) is -250, and +250 in the case of the good decks (i.e., decks C and D). There are two decks with frequent losses (decks A and C), and two decks with infrequent losses (decks B and D). All decks have consistent wins (A and B to have +100, while C and D to have +50) and variable losses (summarized in [15], where scheme 1 [18] has a more variable losses for deck C than scheme 2 [19]).

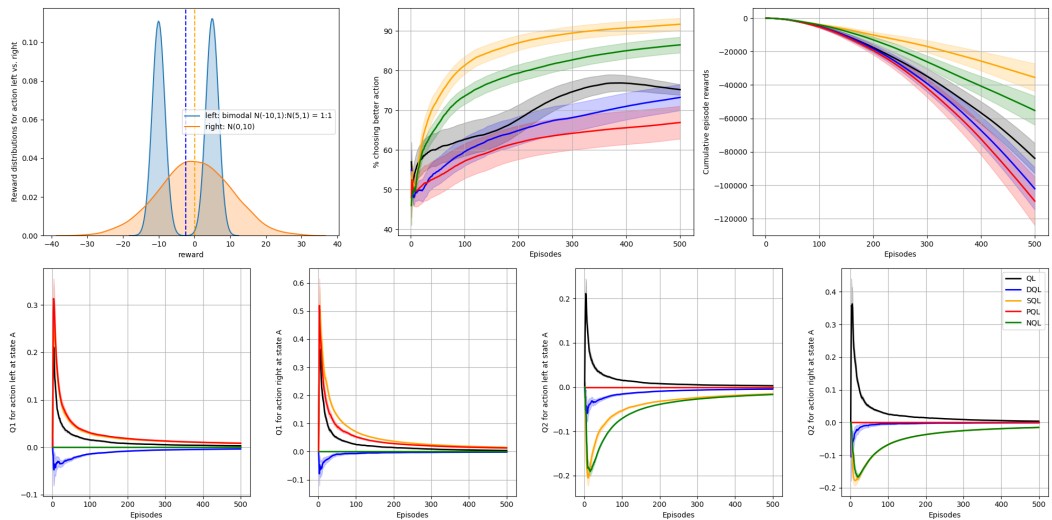

Figure 1: Example bi-modal MDP scenario where HQL performs better than QL and DQL.

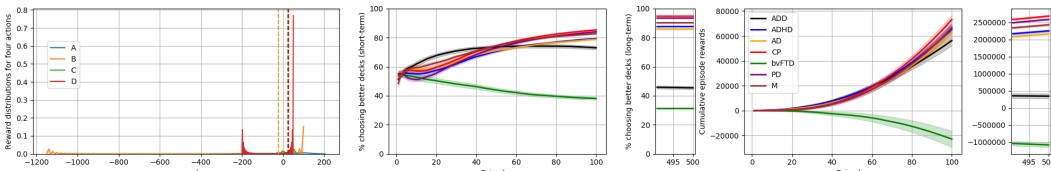

Figure 2: Short-term learning curves of different mental agents in IGT scheme 1.

We performed the each scheme for 100 times over 500 actions. Among the variants of HQL and baselines QL and DQL, CP ("chronic pain") performs best in scheme 1 with the final cumulative rewards of 2689641.25 over 500 draws of cards, followed by NQL (2685174.5) and QL (2673854.75). This is consistent to the clinical implication of chronic pain patients which tend to forget about positive reward information (as modeled by a smaller $\phi_1$) and lack of drive to pursue rewards (as modeled by a smaller $\phi_2$). In scheme 2, SQL performs best with the final score of 2724046.5, followed by NQL (2700618.5) and QL (2689553.5). These examples suggest that the proposed framework has the flexibility to map out different behavior trajectories in real-life decision making (such as IGT). Figure 2 demonstrated the short-term (in 100 actions) and long-term behaviors of different mental agents, which matches clinical discoveries. For instance, ADD ("addiction") quickly learns about the actual values of each decks (as reflected by the short-term curve) but in the long-term still sticks with the decks with a larger wins (despite also with even larger losses). At around 20 actions, ADD performs better than QL and DQL in learning about the decks with the better gains.

## 4 Conclusion

This research proposes a novel parametric family of algorithms for RL problem, extending the classical Q Learning to model a wide range of potential reward processing biases. Our preliminary results support multiple prior observations about reward processing biases in a range of mental disorders, thus indicating the potential of the proposed model and its future extensions to capture reward-processing aspects across various neurological and psychiatric conditions.

MDP Task with 100 randomly generated scenarios of Bi-modal reward distributions

Table 1: Standard agents

|  | QL | DQL | SQL | PQL | NQL |
|---|---|---|---|---|---|
| QL | - | 46 : **54** | 34:**66** | 72 : 28 | 44 : **56** |
| DQL | 54:46 | - | 34:**66** | 59:41 | 50:50 |
| SQL | 66:34 | 66:34 | - | 77:23 | 62:38 |
| PQL | 28:**72** | 41:**59** | 23:**77** | - | 45:**55** |
| NQL | 56:44 | 50:50 | 38:**62** | 55:45 | - |
| avg wins (%) | 0.49 | 0.49 | **0.68** | 0.34 | 0.50 |

Table 2: Mental agents

| SQL |  | ADD | ADHD | AD | CP | bvFTD | PD | M | avg wins (%) |
|---|---|---|---|---|---|---|---|---|---|
| 29:**71** | QL | 60:40 | 65:35 | 73:27 | 43:**57** | 75:25 | 38:**62** | 49:**51** | 0.58 |
| 22:**78** | DQL | 54:46 | 80:20 | 81:19 | 61:39 | 77:23 | 52:48 | 53:47 | 0.65 |
| - | SQL | 78:22 | 94:6 | 95:5 | 67:33 | 89:11 | 66:34 | 81:19 | **0.81** |
| - | avg wins (%) | 0.36 | 0.20 | 0.17 | 0.40 | 0.16 | 0.48 | 0.39 | - |

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

## A    Problem Setting

### A.1    Reinforcement Learning

Reinforcement learning defines a class of algorithms solving problems modeled as a Markov decision process (MDP) [5]. A Markov decision problem is usually denoted by the tuple $(\mathcal{S}, \mathcal{A}, \mathcal{T}, \mathcal{R}, \gamma)$, where $\mathcal{S}$ is a set of possible states, $\mathcal{A}$ is a set of actions ,$\mathcal{T}$ is a transition function defined by ,$\mathcal{T}(s, a, s') = \Pr(s'|s, a)$, where $s, s' \in \mathcal{S}$ and $a \in \mathcal{A}$ ,$\mathcal{R} : \mathcal{S} \times \mathcal{A} \times \mathcal{S} \mapsto \mathbb{R}$ is a reward function , $\gamma$ is a discount factor that specifies how much long term reward is kept.

The goal in an MDP is to maximize the discounted long term reward received. Usually the infinite-horizon objective is considered:

$$\max \sum_{t=0}^{\infty} \gamma^t \mathcal{R}(s_t, a_t, s_{t+1}). \tag{1}$$

Solutions come in the form of policies $\pi : \mathcal{S} \mapsto \mathcal{A}$, which specify what action the agent should take in any given state deterministically or stochastically. One way to solve this problem is through Q-learning with function approximation [20]. The Q-value of a state-action pair, $\mathcal{Q}(s, a)$, is the expected future discounted reward for taking action $a \in \mathcal{A}$ in state $s \in \mathcal{S}$. A common method to handle very large state spaces is to approximate the $\mathcal{Q}$ function as a linear function of some features. Let $\boldsymbol{\psi}(s, a)$ denote relevant features of the state-action pair $\langle s, a \rangle$. Then, we assume $\mathcal{Q}(s, a) = \boldsymbol{\theta} \cdot \boldsymbol{\psi}(s, a)$, where $\boldsymbol{\theta}$ is an unknown vector to be learned by interacting with the environment. Every time the reinforcement learning agent takes action $a$ from state $s$, obtains immediate reward $r$ and reaches new state $s'$, the parameter $\boldsymbol{\theta}$ is updated using

$$\text{difference} = \left[ r + \gamma \max_{a'} \mathcal{Q}(s', a') \right] - \mathcal{Q}(s, a)$$
$$\theta_i \leftarrow \theta_i + \alpha \cdot \text{difference} \cdot \psi_i(s, a), \tag{2}$$

where $\alpha$ is the learning rate. $\epsilon$-greedy is a common strategy used for exploration. That is, during the training phase, a random action is played with a probability of $\epsilon$ and the action with maximum Q-value is played otherwise. The agent follows this strategy and updates the parameter $\boldsymbol{\theta}$ according to Equation (2) until the Q-value converge or for a large number of time-steps.

# B  Related work

In this section, we review prior work in several areas which contributed to the ideas of this paper.

**Reward Processing in Mental Disorders.** The literature on the reward processing abnormalities in particular neurological and psychiatric disorders is quite extensive and fast-growing. We have discussed how dopamine plays a role in reinforcement learning. Parkinson's disease (PD) patients, who have depleted dopamine in the basal ganglia, tend to have impaired performance on tasks that require learning from trial and error. For example, [2] demonstrate that off-medication PD patients are better at learning to avoid choices that lead to negative outcomes than they are at learning to pursue positive outcomes, while dopamine medication used to treat PD symptoms reverses this bias. Alzheimer's disease (AD) is the most common cause of dementia in the elderly and, besides memory impairment, it is associated with executive function impairment and visuospatial impairment. As discussed in [1], AD patients have decreased pursuit of rewarding behaviors, including loss of appetite; these changes are often secondary to apathy, associated with diminished reward system activity. Furthermore, poor performance on certain tasks is correlated with memory impairments. Authors in [21] suggest that the strength of the association between a stimulus and the corresponding response is more susceptible to degradation in Attention-deficit/hyperactivity disorder (ADHD) patients, suggesting impairments in stimulus-response associations. Among other functions, storing the associations requires working memory capacity, which may be impaired in ADHD. In [3], it is demonstrated that patients suffering from addictive behavior have heightened stimulus-response associations, resulting in enhanced reward-seeking behavior for associated stimulus. In [22], chronic pain results in a hypodopaminergic (low dopamine) state that impairs motivated behavior, resulting in reduced reward seeking in these patients. Decreased reward response may underlie a key system mediating the anhedonia and depression, which are common in chronic pain. A variety of computational models was proposed for studying the disorders of reward processing in specific disorders, including, among others [2, 23, 24, 25, 3, 26]. Here, we seek to identify a unifying model that can represent a wide range of reward processing disorders.

**Computational Models of Reward Processing in Mental Disorders.** A wide range of models have been proposed for studying the disorders of reward processing. For example, [2] presented evidence for a mechanistic account of how the human brain implicitly learns to make choices leading to good outcomes, while avoiding those leading to bad ones. Converging evidence in two tasks (a probabilistic one and a deterministic one) in medicated and non-medicated Parkinson's patients has provided support for a dynamic dopamine model of cognitive reinforcement learning. Using a computational multilevel approach, a study presented in [24] suggested that ADHD is associated with impaired gain modulation in systems that generate increased behavioral variability. This computational, multilevel approach to ADHD provides a framework for bridging gaps between descriptions of neuronal activity and behavior, and provides testable predictions about impaired mechanisms. Based on the dopamine hypotheses of cocaine addiction and the assumption of decreased brain reward system sensitivity after long-term drug exposure, the work by [25] proposes a computational model for cocaine addiction. By utilizing average reward temporal difference reinforcement learning, this work incorporates the elevation of basal reward threshold after long-term drug exposure into the model of drug addiction proposed by [3]. The proposed model is consistent with the animal models of drug seeking under punishment. In the case of non-drug reward, the model explains increased impulsivity after long-term drug exposure.

A simple heuristic model developed by [26] simulated individuals' choice behavior by varying the level of decision randomness and the importance given to gains and losses. The findings revealed that risky decision-making seems to be markedly disrupted in patients with chronic pain, probably due to the high cost that pain and negative mood impose on executive control functions. Patients' behavioral performance in decision-making tasks, such as the Iowa Gambling Task (IGT), is characterized by selecting cards more frequently from disadvantageous than from advantageous decks, and by switching more often between competing responses, as compared with healthy controls.

To the best of our knowledge, this work is the first one to propose a generalized version of Reinforcement Learning algorithm which incorporates a range of reward processing biases associated with various mental disorders and shows how different parameter settings of the proposed model lead to behavior mimicking a wide range of impairments in multiple neurological and psychiatric disorders. Most importantly, our reinforcement learning algorithm based on generalization of Q-Learning outperforms the baseline method on multiple artificial scenarios.

## C Reward Processing Models with Different Biases

In this section we describe how specific constraints on the model parameters in the proposed algorithm can yield different reward processing biases discussed earlier, and introduce several instances of the HQL model, with parameter settings reflecting particular biases. The parameter settings are summarized in Table 3, where we use list our models associated with specific disorders.

It is important to underscore that the above models should be viewed as only a first step towards a unifying approach to reward processing disruptions, which requires further extensions, as well as tuning and validation on human subjects. Our main goal is to demonstrate the promise of our parametric approach at capturing certain decision-making biases, as well as its computational advantages over the standard Q-Learning algorithm, due to the increased generality and flexibility facilitated by multi-parametric formulation.

Note that the standard HQL (SQL) approach correspond to setting the four (hyper)parameters used in our model to 1. We also introduce two variants which only learns from one stream of rewards: positive Q-Learning (PQL) and negative Q-Learning (NQL) by setting either $\phi_1, \phi_2$ or $\phi_3, \phi_4$ to zero. Next, we introduce the model which incorporates some mild forgetting of the past rewards or losses, using 0.5 weights, just as an example, and calibrating the other models with respect to this one; we refer to this model as M for "moderate" forgetting, which serves here as a proxy for somewhat "normal" reward processing, without extreme reward-processing biases associated with disorders. We will use the subscript $M$ to denote the parameters of this model.

We will now introduced several models inspired by certain reward-processing biases in a range of mental disorders. *It is important to note that, despite using disorder names for these models, we are not claiming that they provide accurate models of the corresponding disorders, but rather disorder-inspired versions of our general parametric family of models.*

Recall that PD patients are typically better at learning to avoid negative outcomes than at learning to achieve positive outcomes [2]; one way to model this is to over-emphasize negative rewards, by placing a high weight on them, as compared to the reward processing in healthy individuals. Specifically, we will assume the parameter $\phi_4$ for PD patients to be much higher than normal $\phi_4$ (e.g., we use $\phi_4 = 100$ here), while the rest of the parameters will be in the same range for both healthy and PD individuals. Patients with bvFTD are prone to overeating which may represent increased reward representation. To model this impairment in bvFTD patients, the parameter of the model could be modified as follow: $\phi_2^M << \phi_2$ (e.g., $\phi_2 = 100$ as shown in Table 3), where $\phi_2$ is the parameter of the bvFTD model has, and the rest of these parameters are equal to the normal one. To model apathy in patients with Alzheimer's, including downplaying rewards and losses, we will assume that the parameters $\phi_1$ and $\phi_3$ are somewhat smaller than normal, $\phi_1 < \phi_1^M$ and $\phi_3 < \phi_3^M$ (e.g, set to 0.1 in Table 3), which models the tendency to forget both positive and negative rewards. Recall that ADHD may be involve impairments in storing stimulus-response associations. In our ADHD model, the parameters $\phi_1$ and $\phi_3$ are smaller than normal, $\phi_1^M > \phi_1$ and $\phi_3^M > \phi_3$, which models forgetting of both positive and negative rewards. Note that while this model appears similar to Alzheimer's model described above, the forgetting factor will be less pronounced, i.e. the $\phi_1$ and $\phi_3$ parameters are larger than those of the Alzheimer's model (e.g., 0.2 instead of 0.1, as shown in Table 3). As mentioned earlier, addiction is associated with inability to properly forget (positive) stimulus-response associations; we model this by setting the weight on previously accumulated positive reward ("memory") higher than normal, $\tau > \phi_1^M$, e.g. $\phi_1 = 1$, while $\phi_1^M = 0.5$. We model the reduced responsiveness to rewards in chronic pain by setting $\phi_2 < \phi_2^M$ so there is a decrease in the reward representation, and $\phi_3 > \phi_3^M$ so the negative rewards are not forgotten (see table 3).

Of course, the above models should be treated only as first approximations of the reward processing biases in mental disorders, since the actual changes in reward processing are much more complicated, and the parameteric setting must be learned from actual patient data, which is a nontrivial direction for future work. Herein, we simply consider those models as specific variations of our general method, inspired by certain aspects of the corresponding diseases, and focus primarily on the computational aspects of our algorithm, demonstrating that the proposed parametric extension of Q-Learning can learn better than the baseline Q-Learning due to added flexibility.

Table 3: Algorithms Parameters

|  | $\phi_1$ | $\phi_2$ | $\phi_3$ | $\phi_4$ |
|---|---|---|---|---|
| "Addiction" (ADD) | $1 \pm 0.1$ | $1 \pm 0.1$ | $0.5 \pm 0.1$ | $1 \pm 0.1$ |
| "ADHD" | $0.2 \pm 0.1$ | $1 \pm 0.1$ | $0.2 \pm 0.1$ | $1 \pm 0.1$ |
| "Alzheimer's" (AD) | $0.1 \pm 0.1$ | $1 \pm 0.1$ | $0.1 \pm 0.1$ | $1 \pm 0.1$ |
| "Chronic pain" (CP) | $0.5 \pm 0.1$ | $0.5 \pm 0.1$ | $1 \pm 0.1$ | $1 \pm 0.1$ |
| "bvFTD" | $0.5 \pm 0.1$ | $100 \pm 10$ | $0.5 \pm 0.1$ | $1 \pm 0.1$ |
| "Parkinson's" (PD) | $0.5 \pm 0.1$ | $1 \pm 0.1$ | $0.5 \pm 0.1$ | $100 \pm 10$ |
| "moderate" (M) | $0.5 \pm 0.1$ | $1 \pm 0.1$ | $0.5 \pm 0.1$ | $1 \pm 0.1$ |
| Standard HQL (SQL) | 1 | 1 | 1 | 1 |
| Positive HQL (PQL) | 1 | 1 | 0 | 0 |
| Negative HQL (NQL) | 0 | 0 | 1 | 1 |

