# OpenReview forum: "Reinforcement Learning Models of Human Behavior: Reward Processing in Mental Disorders"
_NeurIPS.cc/2019/Workshop/Neuro_AI — Real Neurons & Hidden Units @ NeurIPS 2019 Poster_

### Official Review · AnonReviewer1 · 2019-09-26
**Very interesting idea, but results are not very convincing - and benchmarks may not be suitable for comparison to RL algorithms**

**Clarity:** 2

**Comment:**

The authors propose modeling psychiatric disorders with reinforcement learning, through tracking both a positive as well as a negative q-function. There are presentation issues, and more analyses, tests are needed to convince the reader of the authors claims that that psychiatric disorders can serve a source of inspiration for designing better RL algorithms.


**Category:**

Common question to both AI & Neuro

**Clarity Comment:**

While the motivation of the paper is clear, the method is not explicitly described and requires some digging to understand. Notations are not entirely clear as some deviate from standard RL notations. For instance, important algorithmic details are neglected such as how value tables are updated? Is the task tabular or approximated using deep methods? Was there an eligibility trace? Also, task could be better explained as it is non-obvious to most readers. What are the justifications for comparing with this task, which seems inherently biased to benefit algorithms that  learn multi-modal distributions rather than point estimates? There is also confusion about how the numbers were generated in the end. Also, there is not enough explanations to help the reader understand the figures, especially given that the task is highly specialized and described quickly in words without explanations for the way it’s decided.


**Evaluation:**

3: Good

**Importance:**

3: Important

**Importance Comment:**

This work has important implications for the psychiatric research community, and may be for thinking about reward normalization / reshaping in deep / tabular RL. However, the results are not yet totally convincing as it’s relevant to only one simple task in a tabular setting.


**Intersection:**

3: Medium

**Intersection Comment:**

The authors propose using Q-learning as a framework for modeling individuals with different known reward preferences in psychiatric disorders. The intersection is there, although the authors are drawing connections in specific areas where it’s lacking. For instance, RL can be characterized generally by methods of doing value updates or propagating information about rewards through history. However, the authors are using the framework to examine a very simple, two-choice task.


**Rigor Comment:**

The method proposed is quite simple. There is a need for more experimentation with a wider array of tasks in order to be able to facilitate the author’s claims, since the authors do not fully elucidate the connection to RL in more relevant tasks. It’d be interesting to explore this idea in deep RL with commonly used tasks for the authors to be able to make the claim that they 'outperform state of the art algorithms'.

Overall, the authors seem overly enthusiastic about the prospects of some of the results. The source of the performance gain appears to be possibly from reward normalization / reshaping. While there is a connection, it’s not clear if psychiatric disorders are / should be the source of inspiration for doing better reward normalization / reshaping.

**Technical Rigor:**

2: Marginally convincing

---

### Official Review · AnonReviewer3 · 2019-09-27
**Very interesting work . Definitely deserves a platform for further discussion.**

**Clarity:** 4

**Category:**

AI->Neuro

**Clarity Comment:**

The article has been very well written.

**Evaluation:**

4: Very good

**Importance:**

4: Very important

**Importance Comment:**

This work is important. Task domain should be expanded.

**Intersection:**

3: Medium

**Intersection Comment:**

The article uses refined AI metrics to address neurological disorders. The overall approach could be very rewarding for both fields.

**Rigor Comment:**

The work is convincing to the extend to which one can judge such brief articles.

**Technical Rigor:**

3: Convincing

---

### Official Review · AnonReviewer2 · 2019-09-27
**Innovative ideas in computational psychiatry but quite preliminary results**

**Clarity:** 3

**Comment:**

It would be good to compare and fit the proposed models to real human/primate behavior in normal and pathological conditions and make testable predictions. Also, it would be very interesting to use these models to predict situations that might trigger maladaptive behaviors, by finding scenarios in which the pathological behavior becomes optimal.


**Category:**

AI->Neuro

**Clarity Comment:**

The figures are hard to parse because of the very short captions. One needs to go see Appendix C to understand what the model used (SQL) consists in.

**Evaluation:**

3: Good

**Importance:**

3: Important

**Importance Comment:**

This intriguing study proposes to modify the classical Q-learning paradigm by splitting the reward into two streams with different parameters, one for positive rewards and one for negative rewards. This model allows for more flexibility in modelling human behaviors in normal and pathological states. Although innovative and promising, the work is quite preliminary and would benefit from comparison and validation with real human behavior.

**Intersection:**

3: Medium

**Intersection Comment:**

The work has promising implications for computational psychiatry, but probably not for RL at this point.

**Rigor Comment:**

No comparison with human data.

**Technical Rigor:**

3: Convincing

---

### Decision · Program_Chairs · 2019-10-02

Accept (Poster)